# Association between breastfeeding cessation among under six-month-old infants and postpartum depressive symptoms in Nevada

Smriti Neupane[1], Clariana Vitória Ramos de Oliveira[2], Cláudia Nery Teixeira Palombo[3], Gabriela Buccini[1] *

1 Department of Social and Behavioral Health, School of Public Health, University of Nevada, Las Vegas, Nevada, United States of America, 2 School of Nursing, University of Nevada, Las Vegas, Nevada, United States of America, 3 School of Nursing, Federal University of Bahia, Salvador, Bahia, Brazil

* gabriela.buccini@unlv.edu

**Data Availability Statement:** A de-identified data set is not possible to provide due to ethical considerations due to potentially identifying information in the database. These sharing

## Abstract

### Background

Postpartum depression affects 13% of women after childbirth in the United States. Mothers who experience depression are less likely to breastfeed than those who do not experience depression. On the other hand, breastfeeding may have a positive effect on maternal mental health.

### Research aim

We aimed to analyze whether breastfeeding cessation is associated with postpartum depression symptoms among mothers of infants under six months old in Clark County, Nevada.

### Method

A cross-sectional study was conducted in 2021 using a purposive sample of 305 mother-infant dyads. Postpartum depression symptoms were assessed using the Patient Health Questionnaire-2 (PHQ-2), and the breastfeeding cessation was determined through a 24-hour dietary recall. Descriptive, bivariate, and multivariate logistic regression analyses were conducted.

### Results

Most participants were between 25 and 34 years old (n = 183, 60.0%), multiparous (n = 167, 55.1%), and had a vaginal delivery (n = 204, 70.6%). High frequency of postpartum depressive symptoms was found among mothers who were young (18–24 years) (24.2%), without a partner (25.0%), had unplanned pregnancies (12.7%), and were primiparous (13.2%). Breastfeeding cessation was independently associated with postpartum depressive symptoms (AOR = 3.30, 95% CI: 1.16–9.32) after controlling for sociodemographic, environmental, and obstetric characteristics.

restrictions are imposed by the UNLV Institutional Board Review (IRB). The authors declare that a de-identified data set from this study is available upon request directly to the UNLV IRB (irb@unlv.edu).

**Funding:** The author(s) received no specific funding for this work.

**Competing interests:** The authors have declared that no competing interests exist.

## Conclusion

Breastfeeding cessation is strongly associated with postpartum depressive symptoms among mother-infant dyads in Nevada. Early identification of postpartum depressive symptoms and the promotion of breastfeeding can create a positive feedback loop to foster the well-being of mothers and infants.

## Background

During pregnancy, a woman goes through different hormonal, physical, physiological, and emotional changes, and even after childbirth, she experiences diverse and mixed emotions, from joy and excitement to sadness, fear, lamentation, and anxiety [1]. Many mothers who just delivered a baby can experience 'baby blues', which usually do not affect a mother's capability to take care of a newly born baby and include symptoms like mood swings, worry, sadness, and tiredness that commonly begin within the first 2 to 3 days and resolve in two weeks [1]. However, the persistence and intensity of these symptoms can lead to postpartum depression, which affects the mother's ability to perform her daily routines and take proper care of the child [2]. Postpartum depression occurs after giving birth to an infant, starts soon after the baby's delivery, and lasts up to a year [1]. It influences how mothers feel, act, and think [3, 4]. The clinical symptoms of postpartum depression are feelings of sadness, hopelessness, restlessness, anxiety, irritation, excessive concern, sleep difficulties, problematic decision-making, and suicidal thoughts [2, 3, 5].

Globally, postpartum depression is a common yet serious problem that affects around 15% of women [6]. In the United States (US), 13% of mothers experience postpartum depression after giving birth [2]. Postpartum depression has a critical impact on mothers and poses a long-term risk to their mental health and well-being. Furthermore, it negatively influences the physical, social, and cognitive development of their children [7]. Postpartum depression also impacts the parenting and relationship between the mother and the infant, which eventually affects breastfeeding practices [1, 8].

Breast milk is one of the best sources of nutrition for infants [9]. There is strong evidence that breastfeeding positively impacts the physical, cognitive, and neurological health and well-being of infants, as well as mothers, which translates into economic benefits for society as a whole [10–12]. The American Academy of Pediatrics (AAP) and the World Health Organization (WHO) recommend exclusively breastfeeding an infant for six months and continuing breastfeeding for at least two years or longer if desired [13]. However, less than half of the world's infants are being breastfed as recommended [14], which represents a loss of more than $300 billion each year in unintended benefits from breastfeeding to health and human development [15], when estimating the premature deaths of children due to diarrhea, pneumonia, the occurrence of obesity [16, 17], and other negative outcomes for mothers' health that are demonstrably prevented by breastfeeding [16, 18].

Early breastfeeding cessation refers to a mother who interrupts any type of breastfeeding before an infant completes six months old. On average, the prevalence of any breastfeeding cessation at 6 months in the US is 44.2% [19]. Similarly, sixty percent of the mothers who initiated breastfeeding do not continue breastfeeding as recommended. The common reasons for early cessation are having latching issues, concerns about infant nutrition and weight, illness or using medication, issues with pumping milk, unsupportive work and hospital policies, lack of adequate maternal leave, lack of family support, and cultural barriers [20, 21]. Other risk

factors that diminish breastfeeding practices include low education, first-time pregnancy, lack of prenatal care visits, lack of social support, and postpartum depression [22].

The causal pathways between breastfeeding and postpartum depression can be bidirectional and are still not fully understood [23, 24]. On the one hand, postpartum depression is linked with a greater risk for early breastfeeding cessation [25]. Mothers who experience postpartum depression can face breastfeeding difficulties due to emotional distress and may be less responsive to their infant's feeding cues [26]. As a result, breastfeeding difficulties can lead the mother to feel frustrated and guilty, which exacerbates the risk for postpartum depression symptoms [27]. On the other hand, breastfeeding cessation is linked with a higher risk of postpartum depression [5, 27]. However, the psychological benefits of breastfeeding for the mother are not widely explored and require more research [5, 23, 27].

Better understanding of the factors that impact maternal mental health and depression is important for the health and wellness of mothers and children. Thus, determining factors that are protective against postpartum depression is a high research priority. Our hypothesis is that practicing breastfeeding is associated with maternal mental health. Therefore, the aim of this study is to identify whether breastfeeding cessation is associated with postpartum depressive symptoms.

## Methods

### Study design

A cross-sectional secondary analysis of data from the Early Responsive Nurturing Care (EARN) survey conducted in Clark County, Nevada, in 2021 was conducted. The EARN survey targeted mothers of infants under 6 months living in Clark County, Nevada. The survey consisted of questions regarding socio demographic maternal characteristics, pregnancy, maternal mental health, infant feeding, soothing, and sleeping practices. Ethical approval was provided by the Institutional Review Board of the University of Nevada, Las Vegas, USA (protocol 1767759–2). Maternal consent was obtained before starting the data collection, participation was voluntary and anonymous, and the privacy of the information was maintained.

### Study setting

Clark County is a predominantly urban area located in the southern region of the US state of Nevada. Clark County comprises six jurisdictions–the City of Henderson, the City of Las Vegas, the City of North Las Vegas, Boulder City, the City of Mesquite, and Unincorporated Clark County. As of 2022, the total population of Clark County was 2,350,206, of which 50.2% are female, and 35.6% belong to the age group between 18 and 44. In Clark County, 35.7% of households have an income of less than $49,999, and 7.2% of families with children live below the poverty line [28]. The prevalence of adult depression is higher among females (20.7%) than among males (12.3%) in Clark County, Nevada [28]. Specifically, Nevada's data on the prevalence of postpartum depression after childbirth is not available [29]. The prevalence of any breastfeeding cessation at six months is higher in Nevada (47.5%) than the national average (44.2%) [20].

### Sampling and data collection

The Southern Nevada Health District's (SNHD) 2020 vital records statistics (birth certificates) were the source of the sampling frame for this study. In 2020, 25,604 live births were recorded in the SNHD, and the live births of infants from mothers residing in Clark County were considered the sampling unit. Considering a 95% confidence interval, a 5% error, and assuming

50% completion, the minimum sample size of 268 mothers was estimated. A purposive sampling technique was adopted to recruit mothers for this study.

Only mothers who were 18 years of age or older, had an infant under six months of age, and resided in Clark County (Boulder City, Henderson, Las Vegas, North Las Vegas, and Mesquite), Nevada, were eligible to participate in this study. Similarly, those mothers who were under 18 years of age, had an infant over six months old, and resided outside the Clark County area were excluded from the study. Also, mothers with infants having specific illnesses or needs such as Down syndrome, cleft lip and/or palate, neurological conditions, congenital heart disease, or cardiac problems that prevented or made breastfeeding practices difficult were excluded. Data was collected from August 2021 to October 2021. Mothers were reached through outreach in the community (e.g., distributing flyers in maternal-child care centers, such as prenatal and pediatric offices and daycare centers) and through social media (e.g., Facebook). The majority of the respondents to the survey were enrolled through social media, because in 2021, there were still concerns about the COVID-19 pandemic and in-person interactions were considered unsafe for mothers with infants under 6 months old. The survey was disseminated in Qualtrics [30] and made available in Spanish and English. A total of 323 mothers responded to the survey; however, 18 (5.6%) mothers did not respond to the survey question regarding postpartum depressive symptoms. Thus, the analytical sample of 305 mothers was used to explore the associations between breastfeeding cessation and postpartum depressive symptoms.

## Measurements

**Outcome.** Postpartum depressive symptoms were the main outcome measure for this study. These symptoms were assessed using the Patient Health Questionnaire-2 (PHQ-2) [31], which is a validated tool widely used to screen mothers for depression [32]. It is a two-item instrument that was derived from the PHQ-9 [33]. Each item asked how often the mother has been bothered by the problems over the past two weeks: *little interest or pleasure in doing things;* and *feeling down, depressed, or hopeless*. The responses were recorded as not at all, several days, more than half the days, and nearly every day, based on the seriousness of the situation, and were scored as 0, 1, 2, and 3, respectively. The PHQ-2 score ranged from 0 to 6, and a score of 3 or greater was used as the cutoff to determine if the mothers had postpartum depression symptoms [31].

**Independent variable.** Breastfeeding is a feeding practice indicator defined by the WHO and United Nations Children's Fund (UNICEF) as the act of feeding breast milk to an infant (including milk expressed or from a wet nurse) while the infant may receive any food or liquid including non-human milk and formula [34]. Following this definition, the key independent variable was breastfeeding cessation in the last 24 hours (yes/no). The WHO describes several breastfeeding indicators to measure or study breastfeeding practices among infants and young children during the last 24 hours of the survey to prevent recall bias [11, 34]. Hence, breastfeeding cessation (one of the indicators) information was collected through a 24-hour dietary recall, as recommended by the WHO [34]. The questionnaire asked, 'From yesterday morning until this morning, what has your child eaten?'. The response options were breastmilk, formula or another milk, water/tea/juice, meat/eggs, vegetables, rice/potatoes, beans, sweetened beverages, and others. Breastfeeding cessation was determined if an infant was not fed breast milk within that 24-hour period.

**Covariables.** Covariables were selected based on the conceptual framework [35] and evidence that supported associations with postpartum depression [8, 22, 36]. Study variables were categorized as sociodemographic, environmental, and obstetric characteristics.

Sociodemographic characteristics included maternal age (18–24; 25–34; 35–44), maternal education (primary-secondary-vocational; some college, no degree; associate's, bachelor's, graduate degree), housing situation (owned; rented or others), annual household income (up to $59,999; $60,000–149,000; more than $150,000), marital status (living without a partner; living with a partner), and the number of people living in the household (1–3; more than 4 people).

Environmental characteristics included mothers enrolled in the Special Supplemental Nutrition Program for Women, Infants, and Children (WIC) program (yes; no) and received any public assistance or welfare payments from the government (yes; no). The WIC program is a part of the federal assistance program of the United States that protects and promotes the health of low-income mothers and children up to the age of five, and provides nutritious food, gives information on healthy eating practices, and provides referrals as needed to health care [37]. Obstetric characteristics included planned pregnancy (yes; no), parity (primiparous/multiparous), infant's sex (male; female), co-sleeping (yes; no), mothers with difficulty falling asleep (yes; no), and mothers' sleep hours (less than 5 hours; more than 5 hours), type of delivery (vaginal; c-section), infant separated during the first 24 hours (yes; no), infant admitted in the neonatal unit (yes; no), duration of stay in the maternal ward (1 to 3 days; more than 4 days), and infant breastfed within the first hour (yes; no).

### Data analysis

The Statistical Package for Social Sciences (SPSS), Version 28, was used for the statistical analysis. First, descriptive analysis was conducted to explore the outcome (postpartum depressive symptoms), independent variable (early breastfeeding cessation), and covariables (sociodemographic, environmental, and obstetric characteristics of mothers) using frequencies and percentages. Then, bivariate analysis was computed between postpartum depressive symptoms (a dependent variable) and early breastfeeding cessation (an independent variable), including other covariables, using the chi-square test to explore if there were any correlations among them and also to select the variables to include in the multivariate logistic regression model. All variables with a p-value <0.20 in the bivariate analyses were included in the multivariate logistic regressions. Finally, multivariate logistic regressions were conducted to assess the association of breastfeeding cessation with the postpartum depressive symptoms and estimate the adjusted odds ratios (AOR) at corresponding 95% confidence intervals (CI), after adjusting for confounders. A p-value <0.05 was used as a statistical significance criterion to assess the correlation between the outcome, independent variable, and co-variables.

## Results

A total of 305 mother-infant dyads were included in the analytical sample. Among them, 9.2% of the mothers reported postpartum depressive symptoms. The prevalence of breastfeeding cessation was 26.2%. The majority of the mothers were between 25 and 34 years old (n = 183, 60.0%), had bachelor's or graduate degrees (n = 223, 73.1%), and were not enrolled in the WIC (n = 262, 85.9%). Likewise, most mothers were multiparous (n = 167, 55.1%), had a vaginal delivery (n = 204, 70.6%), and infants were breastfed within the first hour (n = 225, 78.1%) (Table 1).

Young mothers of 18–24 years (n = 8, 24.2%) had a greater prevalence of postpartum depressive symptoms than mothers of the older age group. Similarly, mothers with some college-level education without a degree (n = 10, 20.8%) were more likely to experience postpartum depressive symptoms compared to mothers with primary, secondary or vocational education, and a degree. Mothers without a partner presented a higher frequency of

**Table 1. Descriptive analysis of maternal sociodemographic, environmental, and obstetric characteristics, 2021 (n = 305).**

| Variables | Frequency (n) | Percentage (%) |
|---|---|---|
| **Postpartum Depressive Symptoms** | | |
| Yes | 28 | 9.2 |
| No | 277 | 90.8 |
| **Breastfeeding Cessation** | | |
| Yes | 80 | 26.2 |
| No | 225 | 73.8 |
| **Maternal Age** | | |
| 18–24 | 33 | 10.8 |
| 25–34 | 183 | 60.0 |
| 35–44 | 89 | 29.2 |
| **Maternal Education** | | |
| Primary-Secondary-Vocational | 34 | 11.2 |
| Some college, no degree | 48 | 15.7 |
| Associate's, Bachelor, Graduate Degree | 223 | 73.1 |
| **Housing** | | |
| Owned | 201 | 65.9 |
| Rented or others | 104 | 34.1 |
| **Annual Household Income** | | |
| Up to $59,999 | 79 | 25.8 |
| $60,000- $149,000 | 177 | 58.1 |
| More than $150,000 | 49 | 16.1 |
| **Marital Status** | | |
| Living without partner | 16 | 5.2 |
| Living with partner | 289 | 94.8 |
| **People Living in the Household** | | |
| 1–3 | 124 | 40.7 |
| More than 4 people | 181 | 59.3 |
| **Enrolled in WIC Program** | | |
| Yes | 43 | 14.1 |
| No | 262 | 85.9 |
| **Received any Public Assistance or Welfare Payments from the Government (n = 304)** | | |
| Yes | 37 | 12.2 |
| No | 267 | 87.8 |
| **Planned Pregnancy(n = 303)** | | |
| Yes | 193 | 63.7 |
| No | 110 | 36.3 |
| **Parity (n = 303)** | | |
| Primiparous | 136 | 44.9 |
| Multiparous | 167 | 55.1 |
| **Infant Sex** | | |
| Male | 138 | 45.2 |
| Female | 167 | 54.8 |
| **Co-sleeping (n = 275)** | | |
| Yes | 131 | 47.6 |
| No | 144 | 52.4 |
| **Mother with Difficulty Falling Asleep (n = 275)** | | |

*(Continued)*

**Table 1.** (Continued)

| Variables | Frequency (n) | Percentage (%) |
|---|---|---|
| Yes | 148 | 53.8 |
| No | 127 | 46.2 |
| **Mother's Sleep Hours (n = 275)** | | |
| Less than 5 hours | 68 | 24.7 |
| More than 5 hours | 207 | 75.3 |
| **Type of Delivery (n = 289)** | | |
| Vaginal | 204 | 70.6 |
| C-section | 85 | 29.4 |
| **Infant Separated During First 24 Hour (n = 284)** | | |
| Yes | 145 | 51.1 |
| No | 139 | 48.9 |
| **Infant Admitted in the Neonatal Unit (n = 287)** | | |
| Yes | 35 | 12.2 |
| No | 252 | 87.8 |
| **Duration of Stays in the Maternity Ward (n = 281)** | | |
| 1 to 3 days | 252 | 89.7 |
| More than 4 days | 29 | 10.3 |
| **Infant Breastfed within the First Hour (n = 288)** | | |
| Yes | 225 | 78.1 |
| No | 63 | 21.9 |

experiencing postpartum depressive symptoms (n = 4, 25.0%) compared to those with partners. Mothers with an unplanned pregnancy (n = 14, 12.7%) and having their first child (primiparous) (n = 18, 13.2%) had a higher frequency of experiencing postpartum depressive symptoms compared to mothers with a planned pregnancy and having more than one child (multiparous). In addition, mothers who ceased breastfeeding (n = 13, 16.2%) and those who got less than 5 hours of sleep (n = 9, 13.2%) were more likely to present postpartum depressive symptoms compared to the reference group. Finally, mothers whose infants were admitted to the neonatal unit (n = 6, 17.1%) and not breastfed within the first hour (n = 9, 14.3%) also showed a greater frequency of postpartum depressive symptoms (Table 2).

Multivariate logistic regression revealed that postpartum depressive symptoms are independently associated with early cessation of breastfeeding (AOR = 3.30, 95% CI: 1.16–9.32) after controlling for maternal sociodemographic, environmental, and obstetric characteristic (Table 3).

## Discussion

To our knowledge, this is one of the first studies to assess factors associated with postpartum depressive symptoms in Clark County, Nevada. We found a significant relationship between postpartum depressive symptoms and breastfeeding cessation among six-month-old infants. Mothers who ceased breastfeeding had 3.30 times higher odds of postpartum depressive symptoms compared to mothers who were breastfeeding. Our results are consistent with prior studies that showed an association between breastfeeding and the mental health of mothers, in which non-breastfeeding mothers were at greater risk of depression [8, 22, 36, 38, 39]. A similar study in the United States also concluded that breastfeeding mothers (AOR = 0.87, p = 0.001) had significantly lower risks for depression than non-breastfeeding mothers [40]. Correspondingly, a recent meta-analysis revealed that breastfeeding mothers have a 14%

**Table 2. Prevalence of postpartum depressive symptoms by maternal sociodemographic, environmental, and obstetric characteristics, 2021 (n = 305).** [#].

| Variables | Postpartum Depressive Symptoms | | P-value |
|---|---|---|---|
| | **No** | **Yes** | |
| | **n (%)** | **n (%)** | |
| **Breastfeeding Cessation** | | | 0.01* |
| Yes | 67 (83.75) | 13 (16.25) | |
| No | 210 (93.33) | 15 (6.67) | |
| **Maternal Age** | | | 0.006* |
| 18–24 | 25 (75.76) | 8 (24.24) | |
| 25–34 | 169 (92.35) | 14 (7.65) | |
| 35–44 | 83 (93.26) | 6 (6.74) | |
| **Maternal Education** | | | 0.006* |
| Primary-Secondary-Vocational | 30 (88.24) | 4 (11.76) | |
| Some college, no degree | 38 (79.17) | 10 (20.83) | |
| Associate's, Bachelor, Graduate Degree | 209 (93.72) | 14 (6.28) | |
| **Housing** | | | 0.54 |
| Owned | 184 (91.54) | 17 (8.46) | |
| Rented or others | 93 (89.42) | 11 (10.59) | |
| **Annual Household Income** | | | 0.64 |
| Up to $59,999 | 70 (88.61) | 9 (11.39) | |
| $60,000- $149,000 | 163 (92.09) | 14 (7.91) | |
| More than $150,000 | 44 (89.80) | 5 (10.20) | |
| **Marital Status** | | | 0.02* |
| Living without partner | 12 (75.00) | 4 (25.00) | |
| Living with partner | 265 (91.70) | 24 (8.30) | |
| **People Living in the Household** | | | 0.02* |
| 1–3 | 107 (86.29) | 17 (13.71) | |
| More than 4 people | 170 (93.92) | 11 (6.08) | |
| **Enrolled in WIC Program** | | | 0.54 |
| Yes | 38 (88.37) | 5 (11.63) | |
| No | 239 (91.22) | 23 (8.78) | |
| **Received any Public Assistance or Welfare Payments from the Government (n = 304)** | | | 0.11** |
| Yes | 31 (83.78) | 6 (16.22) | |
| No | 245 (91.76) | 22 (8.24) | |
| **Planned Pregnancy (n = 303)** | | | 0.11** |
| Yes | 179 (92.75) | 14 (7.25) | |
| No | 96 (87.27) | 14 (12.73) | |
| **Parity (n = 303)** | | | 0.03* |
| Primiparous | 118 (86.76) | 18 (13.24) | |
| Multiparous | 157 (94.01) | 10 (5.99) | |
| **Infant Sex** | | | 0.28 |
| Male | 128 (92.75) | 10 (7.25) | |
| Female | 149 (89.22) | 18 (10.78) | |
| **Co-sleeping (n = 275)** | | | 0.49 |
| Yes | 119 (90.84) | 12 (9.16) | |
| No | 134 (93.06) | 10 (6.94) | |
| **Mother with Difficulty Falling Asleep (n = 275)** | | | 0.33 |
| Yes | 134 (90.54) | 14 (9.46) | |
| No | 119 (93.70) | 8 (6.30) | |

*(Continued)*

**Table 2.** (Continued)

| Variables | Postpartum Depressive Symptoms | | P-value |
|---|---|---|---|
| | No | Yes | |
| **Mother Sleep Hours (n = 275)** | | | 0.06** |
| Less than 5 hours | 59 (86.76) | 9 (13.24) | |
| > More than 5 hours | 194 (93.72) | 13 (6.28) | |
| **Type of Delivery (n = 289)** | | | 0.44 |
| Vaginal | 188 (92.16) | 16 (7.84) | |
| C-section | 76 (89.41) | 9 (10.59) | |
| **Infant Separated during First 24 Hours (n = 284)** | | | 0.32 |
| Yes | 131 (90.34) | 14 (9.66) | |
| No | 130 (93.53) | 9 (6.47) | |
| **Infant Admitted in the Neonatal Unit (n = 287)** | | | 0.05* |
| Yes | 29 (82.86) | 6 (17.14) | |
| No | 233 (92.46) | 19 (7.54) | |
| **Duration of Stays in the Maternity Ward (n = 281)** | | | 0.24 |
| 1 to 3 days | 233 (92.46) | 19 (7.54) | |
| More than 4 days | 25 (86.21) | 4 (13.79) | |
| **Infant Breastfed within the First Hour (n = 288)** | | | 0.07** |
| Yes | 209 (92.89) | 16 (7.11) | |
| No | 54 (85.71) | 9 (14.29) | |

*p<0.05

**p<0.20; #total sum (100%) in a row

**Table 3. Multivariate logistic regression on the association between postpartum depressive symptoms and selected maternal sociodemographic, environmental, and obstetric characteristics, 2021.**

| Variables | Unadjusted | | | Adjusted | | |
|---|---|---|---|---|---|---|
| | OR | 95% CI | P>\|z\| | OR | 95% CI | P>\|z\| |
| **Breastfeeding Cessation** | | | | | | |
| Yes | 2.71 | 1.23; 5.99 | 0.01* | 3.30 | 1.16; 9.32 | 0.02* |
| No | 1 | | | 1 | | |
| **Planned Pregnancy** | | | | | | |
| Yes | 1 | | | 1 | | |
| No | 1.86 | 0.85; 4.07 | 0.11 | 1.95 | 0.68; 5.56 | 0.20 |
| **Parity** | | | | | | |
| Primiparous | 1 | | | 1 | | |
| Multiparous | 0.41 | 0.18; 0.93 | 0.03* | 0.89 | 0.08; 8.90 | 0.92 |
| **Infant Admitted in the Neonatal Unit** | | | | | | |
| Yes | 1 | | | 1 | | |
| No | 0.39 | 0.14; 1.06 | 0.06 | 0.46 | 0.1; 1.81 | 0.26 |
| **Mother Sleep Hours** | | | | | | |
| Less than 5 hours | 1 | | | 1 | | |
| >More than 5 hours | 0.43 | 0.17; 1.07 | 0.07 | 0.52 | 0.17; 1.50 | 0.22 |
| **Infant Breastfed within the First Hour** | | | | | | |
| Yes | 1 | | | 1 | | |
| No | 2.17 | 0.91; 5.19 | 0.08 | 1.13 | 0.33; 3.82 | 0.83 |

*p<0.05; Adjusted for maternal age and education, marital status, number of people living in the household, and welfare program.

reduced risk of depression compared to mothers who were not breastfeeding [8]. This might be because breastfeeding mothers are more likely to regulate negative moods and perceived stress, which ultimately prevents them from developing depressive symptoms [41, 42]. Breastfeeding can promote hormonal processes that protect mothers against postpartum depressive symptoms by attenuating cortisol response to stress [22]. Breastfeeding mothers are also more likely to produce calm reactions to stress, which fosters their nurturing behavior [43]. Moreover, evidence indicates that if a mother is unable to breastfeed, it may increase anxiety and depressive symptoms [43]. This is because the social culture of mandatory breastfeeding can make a mother feel responsible for its success or unsuccess [44]. Even though the current study presented strong support for such a relationship, more research is needed to determine the relationship between breastfeeding and postpartum depression outcomes.

Our study did not find an association between postpartum depressive symptoms and maternal characteristics such as planned pregnancy, mother sleep hours, and parity. Corroborating our findings, a longitudinal study in the Netherlands found that women with an unplanned pregnancy reported persistently higher levels of depressive symptoms compared to women with a planned pregnancy [45]. Concerning maternal sleeping hours, similarly to our study researchers found that depressive symptoms during postpartum were associated with poorer sleep, and poorer sleep quality increased postpartum depression symptom severity [46]. On the other hand, the lack of association between parity and postpartum depressive symptoms found in our study differed from the findings from other studies [47, 48]. Martnez-Galiano and colleagues showed that first-time mothers (primiparous) had greater postnatal depressive symptoms and that such mothers were more likely to have issues related to lactation, sadness, and anxiety [47]. On the other hand, Hartmann and colleagues observed an association between having more than one child (multiparous) and greater depression [48]. Overall, it can be inferred that giving birth to a baby, either for the first time or multiple times, can be stressful, and mothers can easily be anxious about how to handle new responsibilities related to their infant. Additionally, the influence of parity on a mother's mental health is poorly studied, and the results are incongruous, highlighting the need to critically analyze and do extensive research on this aspect in the near future.

Infant characteristics such as infant breastfed within the first hour and infant admitted in the neonatal unit were also not associated with postpartum depressive symptoms. Corroborating our findings, in Sweden, pregnant women reported depressive symptoms when they could not accomplish the first breastfeeding session within two hours after birth [49]. Regarding infants admitted to the neonatal intensive unit, a study in Australia showed that postpartum depression was reported within the first 2 weeks after their baby was admitted to the neonatal intensive care unit [50]. Despite the lack of association with postpartum depressive symptoms almost half of the infants were routinely separated from their mothers during the first 24 hours after birth even though very low admission to the neonatal unit was reported. Evidence has demonstrated that routine maternity ward procedures such as early skin-to-skin contact and breastfeeding within the first hours promoted by Baby-Friendly Hospital are protective factors for immediate postpartum depression [51] as well as support maternal role competence [52].

Our results showed that the prevalence of postpartum depressive symptoms (9.2%) among mothers living in Clark County, Nevada, was slightly below the US average (13%) [2]. A recent meta-analysis found increased postpartum depressive symptoms at 6 months postpartum and reported that the prevalence of postpartum depression varied according to country (from 5.0% to 26.32%) [53]. According to their findings, the prevalence of postpartum depression was 8.6% in the US, which is similar to our findings [53]. Our study is the first to date to report the prevalence of postpartum depressive symptoms in Nevada, which is critical to determine the resources and support needed as well as to tailor existing interventions to the needs of the population.

Our findings emphasize the importance of universal mental health screening for postpartum women and appropriate treatment for those with depressive symptoms. Similarly, supportive and evidence-based interventions that address and encourage both breastfeeding and maternal mental well-being should be formulated and implemented. Evidence shows that providing support for mental health well-being may increase breastfeeding practices [54]. Moreover, breastfeeding may protect against or ameliorate depressive symptoms [55]. A systematic review found twelve interventions with a significant positive effect on both maternal mental health and breastfeeding outcomes, including psychoeducational group programs, relaxation therapy, skin-to-skin contact between mother and infant, psychological nursing, motivational interviewing, a health and infant care education program, stepped-care psychological treatment, peer support with home visits, breastfeeding training with home visits, and risk-based treatment with home visits [54]. Thus, health and mental health professionals should inform, educate, and advocate for breastfeeding as it can positively impact mothers' mental health. Most importantly, when delivering these interventions, professionals should take a holistic, integrative approach considering that the impact of depressive symptoms, medication, sleep arrangements, and social support is critical to maternal-child well-being and breastfeeding promotion.

Our study has some limitations that should be considered when interpreting the findings. We surveyed a convenience sample of mothers with infants under six months old in Clark County, Nevada. Data collection efforts across birth, lactation, and pediatric care centers were made to recruit a diverse population of mothers; however, due to the COVID-19 pandemic, most participants were recruited through a paid advertisement on social media. The survey was provided in both English and Spanish, allowing participants to respond to the survey in their preferred language. A comparison of our data to the demographic data (i.e., ethnicity, educational attainment, and household income) in Clark County showed similar trends. Nevertheless, our findings may only be generalized to areas in the US with a similarly high proportion of urban populations as Clark County. Another limitation is that we did not collect the age of the infant at the time of the survey. We acknowledge that some care practices are influenced by the infant's age, and some associations may not be found or weakened due to the lack of the infant age information. Also, we adopted the 24-hour diet recall method as recommended by the WHO to measure breastfeeding cessation to avoid the recall bias. However, if an infant was breastfed before 24 hours or only a couple of days a week, or even never breastfed, it would be classified as breastfeeding cessation. Our questionnaire did not include questions about previous medical diagnosis of depression or other types of psychiatric disorder, and the use of medications such as antidepressants, mood stabilizers, etc. We acknowledge that the lack of this information may underestimate the prevalence of depression symptoms. Finally, because this study was cross-sectional, the causal relationship between postpartum depressive symptoms and breastfeeding cannot be established. In fact, as there was no data regarding postpartum depression in Nevada, our cross-sectional study may establish a baseline for future longitudinal research to clarify this causal relationship. Collectively, our findings contribute to Nevada context-specific hypotheses and support transferability for future public health research exploring the relationship between postpartum depression and breastfeeding [56].

## Conclusion

Our study showed a significant association between postpartum depressive symptoms and breastfeeding cessation. Early identification of postpartum depressive symptoms and the promotion of breastfeeding can create a positive feedback loop to foster the well-being of mothers and infants. More research is needed to clarify the bidirectional causal relationship between postpartum depressive symptoms and breastfeeding cessation.

## Supporting information

**S1 Checklist. STROBE statement—checklist of items that should be included in reports of observational studies.**
(DOCX)

## Author Contributions

**Conceptualization:** Clariana Vitória Ramos de Oliveira, Gabriela Buccini.

**Data curation:** Gabriela Buccini.

**Formal analysis:** Smriti Neupane, Clariana Vitória Ramos de Oliveira, Cláudia Nery Teixeira Palombo.

**Funding acquisition:** Gabriela Buccini.

**Methodology:** Smriti Neupane, Cláudia Nery Teixeira Palombo, Gabriela Buccini.

**Resources:** Gabriela Buccini.

**Supervision:** Gabriela Buccini.

**Writing – original draft:** Smriti Neupane.

**Writing – review & editing:** Clariana Vitória Ramos de Oliveira, Cláudia Nery Teixeira Palombo, Gabriela Buccini.

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
