## [Decision Letter · Decision Letter 0]

3 Aug 2023

PONE-D-23-18811Association between maternal postpartum depressive symptoms and breastfeeding among six-month-old infants in Nevada.PLOS ONE

Dear Dr. Buccini,

Thank you for submitting your manuscript to PLOS ONE. After careful consideration, we feel that it has merit but does not fully meet PLOS ONE’s publication criteria as it currently stands. Therefore, we invite you to submit a revised version of the manuscript that addresses the points raised during the review process.

We look forward to receiving your revised manuscript.

Kind regards,

Rita Amiel Castro

Academic Editor

PLOS ONE

Journal Requirements:

Reviewers' comments:

Reviewer's Responses to Questions

**Comments to the Author**

1. Is the manuscript technically sound, and do the data support the conclusions?

Reviewer #1: Yes

Reviewer #2: Partly

2. Has the statistical analysis been performed appropriately and rigorously? 

Reviewer #1: Yes

Reviewer #2: Yes

3. Have the authors made all data underlying the findings in their manuscript fully available?

Reviewer #1: Yes

Reviewer #2: No

4. Is the manuscript presented in an intelligible fashion and written in standard English?

Reviewer #1: Yes

Reviewer #2: Yes

5. Review Comments to the Author

Reviewer #1: There is no major issue with the manuscript. It is well written and present everything they intended to present. However, there is no new finding arose from this study. Mental health and breastfeeding practices study are well established and the results supported many other previous studies.

Reviewer #2: The manuscript at hand describes a cross-sectional study on the association between breastfeeding and postnatal depression. The authors conducted an online survey in which 305 mothers living in Nevada (U.S.) participated. After controlling for variables, such as sociodemographic, environmental, and obstetric characteristics, the authors found an association between breastfeeding cessation and postnatal depression. The manuscript is generally of interest, but there are a few concerns that should be addressed, before the manuscript can be accepted for publication.

Page 3, line 65 to 66: The authors should state more recent research on the impact of postpartum depression on the development of children. Here, the authors cite Borra et al. (2015). This study is actually not focusing on the association between postpartum depression and infant development.

The authors could cite, e.g.:

Kaplan PS, Danko CM, Everhart KD, et al. Maternal depression and expressive communication in one-year-old infants. Infant Behav Dev. 2014;37(3):398-405.

Murray L, Cooper PJ, Wilson A, Romaniuk H. Controlled trial of the short- and long-term effect of psychological treatment of post-partum depression, 2: impact on the mother-child relationship and child outcome. Br J Psychiatry. 2003;182(5):420-427.

Murray L, Arteche A, Fearon P, Halligan S, Croudace T, Cooper P. The effects of maternal postnatal depression and child sex on academic performance at age 16 years: a developmental approach. J Child Psychol Psychiatry. 2010;51(10):1150-1159.

Schaadt G, Zsido RG, Villringer A, Obrig H, Männel C, Sacher J. Association of Postpartum Maternal Mood With Infant Speech Perception at 2 and 6.5 Months of Age. JAMA Network Open. 2022; 5(9):e2232672.

Page 3, line 69 an following: The authors state that breastfeeding positively impacts society. The authors should elaborate more on this impact, as it might not be obvious at first sight.

Page 4, line 74-75: The authors should elaborate a bit more on the statement that the fact that less than half of the world’s infants are being breastfed as recommended, represents a loss of more than $300 billion each year.

Page 4, line 95 to page 5, line 98: I was a little bit confused that authors state that they hypothesized that practicing breastfeeding can positively influence maternal mental health and reduce maternal postpartum depressive symptoms and that their aim was to identify whether maternal postpartum depressive symptoms are associated with early breastfeeding cessation. The hypothesis, from what I understood, is focusing on the direction of breastfeeding leading to less maternal postpartum mood, while the aim is focusing on the other direction that postpartum depression leads to cessation. As the authors are reporting a cross-sectional study, both cannot really be investigated. I strongly encourage the authors to rephrase the last part of the introduction.

Page 5, line 101: There seems to be a word missing in this sentence.

Page 8, line 167-168: The word “program” should be omitted once, if I am not mistaken.

Page 8, line 179: The authors state that “bivariate analysis between maternal postpartum depressive symptoms was computed across the independent variables”. Here the authors should state why these analyses were performed. At first sight this does not come across.

Table 1: I suggest having the same order for “no” and “yes”. For maternal postpartum depression, the authors first report yes-answers, while they first report no-answers for breastfeeding cessation. This should also be checked for the other variables.

Table 2: I am still not sure whether I understand what the bivariate statistics tell. What exactly did the authors analyze and what does a significant p-value mean. Further, I was confused about the numbers in parenthesis. It is stated that these are percentages. However, the numbers do not add to 100%. Where do the numbers come from? This should be explained or corrected.

Discussion: The authors outline in which way their findings are (mainly) inconsistent with other studies. It would be appropriate to also discuss why these inconsistencies might have occurred. I am not familiar with the sociodemographic characteristics of Nevada or the U.S., but to me the sample did not seem to be entirely representative. I was surprised about the proportion of marital status, as well as annual household income. Maybe this could be one of the reasons for inconsistencies? Independent of the reasons, the authors should elaborate on this more carefully.

Further, I strongly encourage the authors to discuss more deeply on the causality of their results, especially when considering their aim (see comment above). The authors cannot make any conclusions about causality and therefore cannot conclude that breastfeeding reduces the risk of postpartum depression.

6. PLOS authors have the option to publish the peer review history of their article (what does this mean?). If published, this will include your full peer review and any attached files.

Reviewer #1: No

Reviewer #2: No

---

## [Author Response · Author response to Decision Letter 0]

16 Sep 2023

To the Editor and reviewers of PLOS ONE:

Thank you for taking the time to review our manuscript and provide detailed and constructive feedback. We reviewed your comments meticulously and revised them accordingly. We hope they meet your expectations. Please find our point-by-point response below.

Reviewer #1

There is no major issue with the manuscript. It is well written and present everything they intended to present. However, there is no new finding arose from this study. Mental health and breastfeeding practices study are well established and the results supported many other previous studies.

Response: Thank you for this feedback. While our study did not unveil groundbreaking findings, it contributes to discussing the context of large urban areas in the US, specifically in Nevada where maternal mental health and breastfeeding practices are understudied. We included more about the context in the introduction (page 6, lines 100-109) and in the discussion sections (pages 17-18, lines 258-265).

1. The title needs to be changed/rephrased and this study did not measure among six-month-old infants. Please mention ‘breastfeeding cessation’ and ‘under six months’ in the title.

Response: Thank you for specifying this. It has been corrected (page 1, lines 3-5).

2. Line 53 – please clarify the term ‘new mother’. Does it refer to first-time mother?

Response: ‘New mother’ refers to mothers who just delivered a baby, either for the first time or not. It has been clarified (page 4, lines 49-52).

3. Line 86 – statement needs reference/s

Response: Done. (page 5, lines 89).

4. What are the exclusion criteria? Please describe.

Response: Thank you for the comment. We have specified the exclusion criteria (page 8, lines 136-140).

5. As the known prevalence of depression is higher among females, do you include those with mental disorder diagnoses in this study?

Response: Mental health diagnoses were not an exclusion criterion. Thus, everyone who was eligible and wished to participate in the survey could do so, irrespective of their mental health status. (page 8, lines 140-142).

6. Line 157 - please explain more about the 24-hour diet recall used to determine breastfeeding practice. Why this method is chosen? Is 24-hour diet recall accurate to measure breastfeeding cessation? If infants are still fed with breast milk a few times a week, does it consider ‘breastfeeding cessation’?

Response: As recommended by WHO, we used the 24-hour dietary recall method to assess breastfeeding cessation. We clarified the method used and the definition of the breastfeeding cessation indicator in the methods section (page 9, lines 164-176). In addition, we added a discussion of the potential bias in the discussion section (pages 20-21, lines 328-333).

7. Line 167 – please describe WIC

Response: A brief description has been added (page 10, lines 187-190).

8. Line 196 – Please rearrange the paragraph according to the variables in Table 2. The current arrangement makes it harder to refer to the table.

Response: Thank you for the suggestion. It has been rearranged. (page 13-14, lines 221-233).

9. Table 2: Some categories include only very few participants. Even though the result is significant if only three participants are included in the category, how marked the result is? Please kindly discuss this.

Response: We have rectified the mentioned categories and thoroughly reevaluated the analysis (Table 2, pages 14-15).

10. Relationship between postpartum mental health and breastfeeding and well-versed. What is the difference with your study?

Response: Thank you for this feedback. While our study did not unveil groundbreaking findings, it contributes to discussing the context of large urban areas in the US, specifically in Nevada where maternal mental health and breastfeeding practices are understudied. We included more about the context in the introduction (page 6, lines 100-109) and in the discussion sections (pages 17-18, lines 258-265).

Reviewer #2: 

The manuscript at hand describes a cross-sectional study on the association between breastfeeding and postnatal depression. The authors conducted an online survey in which 305 mothers living in Nevada (U.S.) participated. After controlling for variables, such as sociodemographic, environmental, and obstetric characteristics, the authors found an association between breastfeeding cessation and postnatal depression. The manuscript is generally of interest, but there are a few concerns that should be addressed, before the manuscript can be accepted for publication.

Response: Thank you for this feedback. We appreciate your detailed notes to help to improve the manuscript.

Page 3, line 65 to 66: The authors should state more recent research on the impact of postpartum depression on the development of children. Here, the authors cite Borra et al. (2015). This study is actually not focusing on the association between postpartum depression and infant development.

The authors could cite, e.g.:

Kaplan PS, Danko CM, Everhart KD, et al. Maternal depression and expressive communication in one-year-old infants. Infant Behav Dev. 2014;37(3):398-405.

Murray L, Cooper PJ, Wilson A, Romaniuk H. Controlled trial of the short- and long-term effect of psychological treatment of post-partum depression, 2: impact on the mother-child relationship and child outcome. Br J Psychiatry. 2003;182(5):420-427.

Murray L, Arteche A, Fearon P, Halligan S, Croudace T, Cooper P. The effects of maternal postnatal depression and child sex on academic performance at age 16 years: a developmental approach. J Child Psychol Psychiatry. 2010;51(10):1150-1159.

Schaadt G, Zsido RG, Villringer A, Obrig H, Männel C, Sacher J. Association of Postpartum Maternal Mood With Infant Speech Perception at 2 and 6.5 Months of Age. JAMA Network Open. 2022; 5(9):e2232672.

Response: Thank you for pointing this out. We added the reference (page 4, line 64). 

Page 3, line 69 a following: The authors state that breastfeeding positively impacts society. The authors should elaborate more on this impact, as it might not be obvious at first sight.

Response: We restructured the sentence including a new reference (page 4, lines 67-70).

Page 4, line 74-75: The authors should elaborate a bit more on the statement that the fact that less than half of the world’s infants are being breastfed as recommended, represents a loss of more than $300 billion each year.

Response: We included new references to support the statement (page 5, lines 72-77).

Page 4, line 95 to page 5, line 98: I was a little bit confused that authors state that they hypothesized that practicing breastfeeding can positively influence maternal mental health and reduce maternal postpartum depressive symptoms and that their aim was to identify whether maternal postpartum depressive symptoms are associated with early breastfeeding cessation. The hypothesis, from what I understood, is focusing on the direction of breastfeeding leading to less maternal postpartum mood, while the aim is focusing on the other direction that postpartum depression leads to cessation. As the authors are reporting a cross-sectional study, both cannot really be investigated. I strongly encourage the authors to rephrase the last part of the introduction.

Response: We corrected the hypothesis and rephrased the last part of the introduction (page 6, lines 103-105).

Page 5, line 101: There seems to be a word missing in this sentence.

Response: We have corrected the sentence (page 7, lines 112-113).

Page 8, line 167-168: The word “program” should be omitted once, if I am not mistaken.

Response: Done (page 10, lines 185-187).

Page 8, line 179: The authors state that “bivariate analysis between maternal postpartum depressive symptoms was computed across the independent variables”. Here the authors should state why these analyses were performed. At first sight this does not come across.

Response: We have explained why bivariate analysis was done (page 11, lines 202-206).

Table 1: I suggest having the same order for “no” and “yes”. For maternal postpartum depression, the authors first report yes-answers, while they first report no-answers for breastfeeding cessation. This should also be checked for the other variables.

Response: Done. We have checked with other variables and updated the tables accordingly (Table 1, page 12).

Table 2: I am still not sure whether I understand what the bivariate statistics tell. What exactly did the authors analyze and what does a significant p-value mean. Further, I was confused about the numbers in parenthesis. It is stated that these are percentages. However, the numbers do not add to 100%. Where do the numbers come from? This should be explained or corrected.

Response: Bivariate analysis is a statistical method to determine if there is a statistical association between the two variables and, if so, how strong and in which direction that association is (reference). Our bivariate analysis is presented in Table 2. First, the cross-tabulation helped to clarify the existence and the direction of an association between maternal postpartum depression outcome, the independent variable, and covariables, individually. For example, among mothers with postpartum depression, 16.25% had ceased breastfeeding compared to 6.67% who did not cease breastfeeding; on the other hand, among mothers without postpartum depression, 83.75% had ceased breastfeeding, while 93.3% did not cease breastfeeding. This description illustrates the direction of a possible association (without any cause-and-effect relationship), which in this case, shows the presence of maternal mental health increases the frequency of breastfeeding cessation. Second, the goal of the chi-square test was to explore how strong the association between maternal postpartum depression and covariables was individual. All covariables with a p-value <0.20 in the bivariate analysis were included in the multivariate logistic regression model.

Reference:

Bertani, A., Di Paola, G., Russo, E., & Tuzzolino, F. (2018). How to describe bivariate data. Journal of thoracic disease, 10(2), 1133–1137. https://doi.org/10.21037/jtd.2018.01.134

Discussion: The authors outline in which way their findings are (mainly) inconsistent with other studies. It would be appropriate to also discuss why these inconsistencies might have occurred. I am not familiar with the sociodemographic characteristics of Nevada or the U.S., but to me the sample did not seem to be entirely representative. I was surprised about the proportion of marital status, as well as annual household income. Maybe this could be one of the reasons for inconsistencies? Independent of the reasons, the authors should elaborate on this more carefully.

Response: Thank you for this note. We added a paragraph discussing the hypothesis of the different findings from our study to other studies as well as how it relates to the Nevada context (pages 19-20, lines 305-317).

Further, I strongly encourage the authors to discuss more deeply on the causality of their results, especially when considering their aim (see comment above). The authors cannot make any conclusions about causality and therefore cannot conclude that breastfeeding reduces the risk of postpartum depression.

Response: We agree with your comment. Following the suggestions, we reframed the aim of the study in the background section (page 6, lines 107-109). We added a paragraph expanding on the discussion of the limitations of our study due to sampling and design (page 18, lines 274-279, page 19, lines 300-304, & page 20, lines 318-322). We also redid the conclusion (page 21, lines 335-346).

---

## [Decision Letter · Decision Letter 1]

5 Dec 2023

PONE-D-23-18811R1Association between maternal postpartum depressive symptoms and breastfeeding cessation among under six-month-old infants in Nevada.PLOS ONE

Dear Dr. Buccini,

Thank you for submitting your manuscript to PLOS ONE. After careful consideration, we feel that it has merit but does not fully meet PLOS ONE’s publication criteria as it currently stands. Therefore, we invite you to submit a revised version of the manuscript that addresses the points raised during the review process.

We look forward to receiving your revised manuscript.

Kind regards,

Rita Amiel Castro

Academic Editor

PLOS ONE

Journal Requirements:

Reviewers' comments:

Reviewer's Responses to Questions

**Comments to the Author**

1. If the authors have adequately addressed your comments raised in a previous round of review and you feel that this manuscript is now acceptable for publication, you may indicate that here to bypass the “Comments to the Author” section, enter your conflict of interest statement in the “Confidential to Editor” section, and submit your "Accept" recommendation.

Reviewer #2: (No Response)

Reviewer #3: (No Response)

Reviewer #4: (No Response)

2. Is the manuscript technically sound, and do the data support the conclusions?

Reviewer #2: Yes

Reviewer #3: Partly

Reviewer #4: Yes

3. Has the statistical analysis been performed appropriately and rigorously? 

Reviewer #2: Yes

Reviewer #3: Yes

Reviewer #4: I Don't Know

4. Have the authors made all data underlying the findings in their manuscript fully available?

Reviewer #2: No

Reviewer #3: Yes

Reviewer #4: No

5. Is the manuscript presented in an intelligible fashion and written in standard English?

Reviewer #2: Yes

Reviewer #3: Yes

Reviewer #4: Yes

6. Review Comments to the Author

Reviewer #2: The manuscript at hand is a study I reviewed earlier. The authors have done a good job in revising the manuscript and responding to the reviewer comments.

I have one minor comment left and can then suggest the manuscript for publication.

In my previous comments, I asked the authors to explain why they performed the bivariate statistics. Further, I pointed out that in Table 2, the numbers do not add to 100%. In their revised manuscript the authors explained the aim of the bivariate statistics more precisely and in their response, the authors explained why the numbers in table 2 do not add up to 100%. However, I could not find any information in the manuscript and would strongly suggest to add a note to table 2, to explain the percentage more precisely. If I overread this information, please feel free to ignore this comment.

Reviewer #3: Thank you for the well-written manuscript. However, although I acknowledge that there are no estimates for postpartum depression symptoms, the study was conducted with a small sample using purposive sampling and a cross-sectional study design, so this manuscript does not contribute new knowledge to the field.

Reviewer #4: You provide these two statements: Line 27 (abstract): “We aimed to analyze whether maternal postpartum depression symptoms are associated with early breastfeeding cessation”; Line 108 (background): “Thus, we aimed to find out whether early breastfeeding cessation is an independent risk factor for maternal postpartum depression”. You should be clearer about the aim of the study/ in which direction are you seeking an association.

Line 171: A 24h dietary recall is not the best method, since a mother could need to transiently interrupt breastfeeding due to medication/medical exams and resume again later. You explain this in the discussion, however, if a mother was not breastfeeding, how do you know when she stopped? If you don’t have this information, you should state it as a limitation of your study.

Besides, did you have the option “breastmilk + formula” in your questionnaire?

Line 177: Regarding the covariables, did you ask the mothers if they had a diagnosis of depression/other psychiatric disorder and/or if they were taking any medication? It is important to know this as some treatments are not compatible with breastfeeding.

Table 1: I suggest you write “postpartum depression symptoms” instead of “maternal postpartum depression” in your table, since you do not have a clinical diagnosis.

Table 1: There is a high prevalence of newborns separated from their mothers in the first hour, yet a low prevalence of admissions in the neonatal unit. While it is not the primary goal of the study, this finding deserves some discussion.

Table 2: You should not write in the title “risk for” since you did not evaluate causality.

Line 305: there is a grammatical error.

Overall, the study is well written and gives information about the prevalence of postpartum depression symptoms and mothers who did not breastfeed in the previous 24h. While this is useful information and the study has a good sample size, there are some variables that were not taken into account: there is no information regarding a diagnosis of depression or medication taken by mothers, no information about the timing of breastfeeding cessation or if there were mothers who never breastfed. These are important limitations and should be mentioned.

7. PLOS authors have the option to publish the peer review history of their article (what does this mean?). If published, this will include your full peer review and any attached files.

Reviewer #2: No

Reviewer #3: No

Reviewer #4: No

---

## [Author Response · Author response to Decision Letter 1]

21 Dec 2023

To the Editor and reviewers of PLOS ONE:

Thank you for taking the time to review our manuscript and provide detailed and constructive feedback. We reviewed your comments meticulously and revised them accordingly. We hope they meet your expectations. Please find our point-by-point response below.

Reviewer #2: The manuscript at hand is a study I reviewed earlier. The authors have done a good job in revising the manuscript and responding to the reviewer comments.

I have one minor comment left and can then suggest the manuscript for publication.

In my previous comments, I asked the authors to explain why they performed the bivariate statistics. Further, I pointed out that in Table 2, the numbers do not add to 100%. In their revised manuscript the authors explained the aim of the bivariate statistics more precisely and in their response, the authors explained why the numbers in table 2 do not add up to 100%. However, I could not find any information in the manuscript and would strongly suggest adding a note to table 2, to explain the percentage more precisely. If I overread this information, please feel free to ignore this comment.

Response: Thank you for your feedback. We added a footnote in Table 2 explaining 100% is the sum of the numbers in a row (page 14, line 215).

Reviewer #3: Thank you for the well-written manuscript. However, although I acknowledge that there are no estimates for postpartum depression symptoms, the study was conducted with a small sample using purposive sampling and a cross-sectional study design, so this manuscript does not contribute new knowledge to the field.

Response: Thank you for the opportunity to reflect on your contribution to new knowledge in the field. Our team believes that research should always seek to strike a balance between (1) context-specific knowledge as well as (2) transferability. Our study contributes to the field by being the first to report the prevalence of postpartum depressive symptoms and investigate the independent association with breastfeeding cessation in Nevada (context). Furthermore, while corroborating existing literature, it also invites readers of the research to make connections between elements of our study and their own experiences (transferability). Collectively, our findings contribute to context-specific hypotheses and support transferability for future public health research exploring the relationship between postpartum depression and breastfeeding. Therefore, we believe that this manuscript, despite its limitations (detailed in the discussion section), makes a valuable contribution to the literature and the field of public health.

Reviewer #4: You provide these two statements: Line 27 (abstract): “We aimed to analyze whether maternal postpartum depression symptoms are associated with early breastfeeding cessation”; Line 108 (background): “Thus, we aimed to find out whether early breastfeeding cessation is an independent risk factor for maternal postpartum depression”. You should be clearer about the aim of the study/ in which direction are you seeking an association.

Response: Thank you for pointing this out. We now have stated our aim in a consistent way or direction: ‘To analyze whether breastfeeding cessation is associated with postpartum depression among mothers of infants under six months old in Clark County, Nevada (both in title, abstract (lines 26–27), and introduction (lines 91–92)).

Line 171: A 24h dietary recall is not the best method, since a mother could need to transiently interrupt breastfeeding due to medication/medical exams and resume again later. You explain this in the discussion, however, if a mother was not breastfeeding, how do you know when she stopped? If you don’t have this information, you should state it as a limitation of your study.

Response: Thank you for this comment. Specifically, we do not have information on when the mother stopped breastfeeding and have included it in the limitation paragraph. While we acknowledge the potential limitations of a 24-hour dietary recall, we used it to standardize measurement when assessing infant feeding outcomes and to avoid recall bias as recommended by the WHO. It also allows future comparisons across studies and populations. We added a paragraph in the discussion section to discuss the limitations pointed out by the reviewer (Page 18, lines 304–309).

Besides, did you have the option “breastmilk + formula” in your questionnaire?

Response: We did ask about mixed feeding with ‘breastmilk+formula’ in the 24-hour dietary recall. Because our independent variable refers to any breastfeeding, in the case of the affirmative response, this dyad was classified as breastfeeding (Page 7, lines 154-157).

Line 177: Regarding the covariables, did you ask the mothers if they had a diagnosis of depression/other psychiatric disorder and/or if they were taking any medication? It is important to know this as some treatments are not compatible with breastfeeding.

Response: We do acknowledge the importance of these questions; however, unfortunately, they were not included in the questionnaire. We added a paragraph in the discussion section to discuss the limitations pointed out by the reviewer (Page 18, lines 307-309).

Table 1: I suggest you write “postpartum depression symptoms” instead of “maternal postpartum depression” in your table, since you do not have a clinical diagnosis.

Response: We standardize “postpartum depressive symptoms” throughout the manuscript. It also has been corrected in Table 1.

Table 1: There is a high prevalence of newborns separated from their mothers in the first hour, yet a low prevalence of admissions in the neonatal unit. While it is not the primary goal of the study, this finding deserves some discussion.

Response: Thank you for this feedback. We added a paragraph to discuss this important insight in our discussion (Page 16, lines 259 - 269).

Table 2: You should not write in the title “risk for” since you did not evaluate causality.

Response: Thank you for your suggestion. We removed it throughout the manuscript.

Line 305: there is a grammatical error.

Response: Thank you for noticing this. It has been corrected (Pages 20, line 312-316).

Overall, the study is well written and gives information about the prevalence of postpartum depression symptoms and mothers who did not breastfeed in the previous 24h. While this is useful information and the study has a good sample size, there are some variables that were not taken into account: there is no information regarding a diagnosis of depression or medication taken by mothers, no information about the timing of breastfeeding cessation or if there were mothers who never breastfed. These are important limitations and should be mentioned.

Response: Thank you for your feedback. We updated this information in our discussion, where we specified our limitations (Page 18, lines 304-309).

---

## [Editor Report · Decision Letter 2]

2 Jan 2024

Association between breastfeeding cessation among under six-month-old infants and postpartum depressive symptoms in Nevada.

PONE-D-23-18811R2

Dear Dr. Buccini,

We’re pleased to inform you that your manuscript has been judged scientifically suitable for publication and will be formally accepted for publication once it meets all outstanding technical requirements.

Kind regards,

Rita Amiel Castro

Academic Editor

PLOS ONE
---

## [Editor Report · Acceptance letter]

19 Jan 2024

PONE-D-23-18811R2 

PLOS ONE

Dear Dr. Buccini, 

I'm pleased to inform you that your manuscript has been deemed suitable for publication in PLOS ONE. Congratulations! Your manuscript is now being handed over to our production team.

Kind regards, 

on behalf of

Dr. Rita Amiel Castro 

Academic Editor

PLOS ONE